

# Suitable oasis scales under a government plan in the Kaidu-Konqi River Basin of northwest arid region, China

Aihong Fu, Weihong Li, Yaning Chen and Yuting Liu

State Key Laboratory of Desert and Oasis Ecology, Xinjiang Institute of Ecology and Geography, Chinese Academy of Sciences, Urumqi, Xinjiang, China

## ABSTRACT

The Yanqi Basin and the Konqi River Basin of the Kaidu-Konqi River Basin were chosen as the study sites in this paper in order to investigate suitable scales of natural and artificial oases with a specified water resource and water quantity planned by the local government. Combined with remote-sensing images from 2013, water resources in 2013, 2025 and 2035, and weather and socioeconomic data, suitable scales of oases were analyzed. The results showed that: (1) The total available water quantities in the Yanqi Basin and the Konqi River Basin without river base flow, and the input of water into Bosten Lake and Tarim River, over high-, normal and low-flow periods, in 2025 and 2035, were $19.04 \times 10^8$ m$^3$, $10.52 \times 10^8$ m$^3$, $4.95 \times 10^8$ m$^3$, $9.95 \times 10^8$ m$^3$ and $9.95 \times 10^8$ m$^3$, as well as $21.77 \times 10^8$ m$^3$, $13.95 \times 10^8$ m$^3$, $10.11 \times 10^8$ m$^3$, $12.50 \times 10^8$ m$^3$, and $9.74 \times 10^8$ m$^3$. (2) The water demand of the natural oasis in the Yanqi Basin and the Konqi River Basin was $2.59 \times 10^8$ m$^3$, and $4.59 \times 10^8$ m$^3$, respectively. (3) The total water consumption of the artificial oasis in 2013, 2025, and 2035 were $10.51 \times 10^8$ m$^3$, $10.99 \times 10^8$ m$^3$ and $10.74 \times 10^8$ m$^3$ in the Yanqi Basin, respectively, and $18.59 \times 10^8$ m$^3$, $14.07 \times 10^8$ m$^3$ and $13.30 \times 10^8$ m$^3$ in the Kongqi River Basin, respectively. (4) Under government planning, the optimal area in 2025 and 2035 should be 5,100.06 km$^2$ and 5,096.15 km$^2$ in the Yanqi Basin oases, and 6,008.53 km$^2$ and 4,691.36 km$^2$ in the Konqi River Basin oases, respectively, under the different inflow variations, and 4,972.71 km$^2$ and 4,969.22 km$^2$ in the Yanqi Basin oases, and 5,975.17 km$^2$ and 4,665.67 km$^2$ in the Kongqi River Basin oases, respectively, under the appropriate proportion. (5) The artificial oases in these basins should be greatly decreased in the future due to limited water resources.

Corresponding author
Weihong Li, liwh@ms.xjb.ac.cn

## INTRODUCTION

Mountain-oasis-desert is the main topographic feature of inland river basins in China (*Sun, Wang & Yang, 2007*). Oases, major human activity and economic development zones, exist in a desert background, and constitute an essential part of arid and semi-arid regions (*Su et al., 2007*). Their scale, location, and development depend on the carrying capacity of water resources (*Wang et al., 2011; Moharram et al., 2012*). Water resource originates from the mountain area located in the upper reaches of the basin, and is mainly composed of

mountain precipitation and ice snow melt water (*Chen et al., 2016*). The flow at mountain-pass supplies water consumption of natural and artificial oases (*Huang et al., 2011*). Therefore, river runoff is one of the very important factors affecting oasis development (*Li et al., 2015*). However, mountain precipitation, ice snow melt water, and river runoff are affected by climate change and variability (*Coppola, Raffaele & Giorgi, 2016*). It has been reported that climate warming has led to the decrease of mountain solid precipitation, the increase of ice snow melt water and the temporary increase of river runoff (*Piao et al., 2010*; *Chen et al., 2016*), but the water resources in the mountain area decrease in the long run if the temperature continues to rise (*Piao et al., 2010*; *Chen et al., 2016*). The decrease of water resources will seriously affect oasis development (*Piao et al., 2010*), and intensify the shortage of water resources and the contradiction between supply and demand because of the increase of population, resulting in the unsustainable development of oases. Realizing the stable and sustainable of oasis, the local government has planned the future supply of water resources based on the status of river runoff and the development scale of oases in order to relieve the shortage of water resources and ensure the healthy development of oases (*Crabbe & Robin, 2006*). What scale of oasis development can be met by the future quantity of water resources planned by the local government is a critical issue yet to be addressed.

The suitable scale of oases has recently attracted the attention of researchers (*Lei et al., 2006*; *Fan et al., 2000*; *Maneta et al., 2009*; *Contreras et al., 2011*), but studies focusing on continental river basins of arid regions remain scarce in China. Of the few studies reported, most have focused on calculating the suitable scale of oases in the present year on the basis of runoff, ecological water demand of natural oases, and water demand of artificial oases (*Lei, Li & Ling, 2015*; *Guo et al., 2016*; *Huang, Shen & Zhang, 2008*; *Cao et al., 2012*; *Ling et al., 2012*; *Zheng et al., 2011*; *Hu et al., 2006*). However, the extant literature has rarely involved the prediction of suitable scales of oases in the future, especially when the allocation and use of water resources are planned by local governments. Only *Lei, Li & Ling (2015)* predicted the suitable size of the Keriya River Basin, Xinjiang, in 2020, according to fitting models between surface runoff and climate factors; however, the regulatory role of the government was not taken into account. Government planning of water resource supply and demand constitutes an essential factor in arid regions, especially in arid areas where water resources are limited and water conflicts are prominent. Because government limitations on the supply and demand of water resources seriously constrain oasis development, it is very important to investigate the impact of government planning on oasis development. This type of investigation would provide a scientific basis for policy-makers to coordinate the development scale of different departments and ecological, economic, and social water use in oases.

The Kaidu-Konqi River Basin (40.00–43.20N, 82.55–90.15E) is located in the central section of the Xinjiang Uygur Autonomous Region, China, in the north portion of the Tarim Basin. The total area of the Kaidu-Konqi River Basin is 77,260 km$^2$ and encompasses the Kaidu River, Bosten Lake, and the Konqi River. The Kaidu River originates in the south of the middle Tianshan Mountains and flows through Bosten Lake into the Konqi River. The administrative divisions comprise six counties (cities) (i.e., Yanqi, Hejing, Heshuo, Bohu, Korla, and Yuli), 11 military regiment production-construction farms, four state farms,

and oil enterprises belonging to the Mongolia Autonomous Prefecture of Bayinguoleng. Kaidu River is a river mixed with snow, ice, and rainwater with an annual mean runoff of $35.05 \times 10^8$ m$^3$. There are abundant surface and ground water resources, with total mean available water resources of $42.00 \times 10^8$ m$^3$. However, as a consequence of the rapid expansion of artificial oases since the 1950s, water consumption in artificial oases has mostly increased, which has greatly crowded out ecological water so that desert riparian forests located in the lower reaches of the Kaidu-Konqi River Basin have substantially declined. To alleviate the contradiction between supply and demand of water resources, and ensure the healthy development of oasis, lake and desert ecosystems, the local government formulated plans for the supply and demand of water resources in the Kaidu-Konqi River Basin in 2025 and 2035. Based on this government plan, determining precisely how water resources can be optimally allocated and the most suitable scale of oases in these basins need to be elucidated.

Based on previous research, the present study took into account the government plan for the limitation of the supply and demand of oasis water resources, including collected water resource data (1958–2013), meteorological data (2013), socio-economic data (2013), and Landsat TM remote-sensing images (2013) with the resolution ratio of 30 m, to analyze high- and low-flow variations of surface runoff in the Kaidu River and the Konqi River. The required quantities of water resources were calculated for maintaining the socioeconomic development of artificial oases in the Kaidu-Konqi River Basin. We calculated the required ecological water quantities of natural vegetation. On the basis of the above analysis, we also discussed the suitable scale of natural and artificial oasis in 2013, as well as in 2025 and 2035 under the government plan. This study aimed to provide a sound theoretical basis for determining reasonable oasis development management plans.

## MATERIALS & METHODS

### Study region

The Kaidu-Konqi River Basin is located in the Northeastern Tarim River and the Northeastern margin of the Taklimakan Desert. The total area of the Kaidu-Konqi River Basin is 77,260 km$^2$, with a mountainous area of 34,660 km$^2$ and a plain area of 42,600 km$^2$. The Kaidu River originates in the Yilianhabierga Mountains in the center of the south of the Tianshan Mountains, with a length of 560 km. It flows through the Yanqi Basin and into Bosten Lake. The outflow of Bosten Lake is called the Konqi River. The Kaidu River consists of ice and snow melt, with a length of 560 km and an average annual runoff of $35.05 \times 10^8$ m$^3$. The Konqi River originates from Bosten Lake, with a length of 942 km and an average annual runoff of $13.34 \times 10^8$ m$^3$.

The Kaidu-Konqi River Basin lies in the hinterland of Eurasia, and experiences the extreme desert climate of a warm temperate zone. This basin encompasses a typical mountain-oasis-desert ecosystem in an arid region. The climate varies with the gradual decrease of elevation from mountains, plain oasis, to desert. The mountainous area has an average annual temperature of $-4.3$ °C C, average sunshine hours of 2,789.5 h, average annual precipitation of 275 mm, and evaporation of 680 mm. The oasis in Yanqi Basin has an

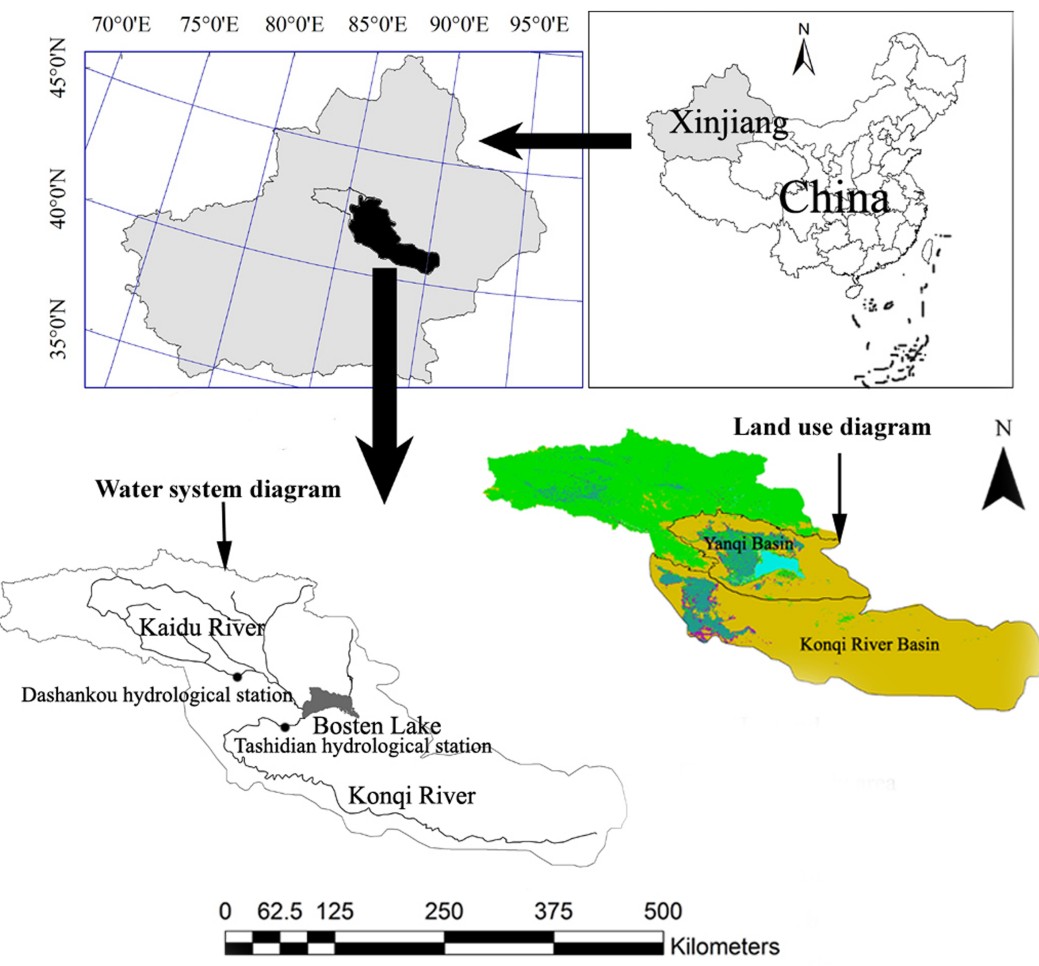

**Figure 1 Location of the study region in Xinjiang, China.** The study region includes the Yanqi Basin and the Konqi River Basin without the mountainous area of the Kaidu-Konqi River Basin.

average annual temperature of 8.3 °C, average sunshine hours of 3,047.6 h, average relative humidity of 57%, average annual precipitation of 76.3 mm, and average annual evaporation of 1,073 mm. The climate in the Konqi River Basin Oasis is drier than in the Yanqi Basin Oasis, with an average annual temperature of 11.0 °C, average annual precipitation of 45.9 mm, average annual evaporation of 1,429 mm, and average sunshine hours of 2,987 h.

In this study, the Yanqi Basin and the Konqi River Oasis were chosen as the study area (Fig. 1). Figure 1 presents land use/cover in 2013 and drainage map in the study areas. The mountainous area of the higher reaches of the Kaidu River Basin cannot be taken into consideration because water resources originated from the mountains, and mountain-pass water resources cannot be used by mountain vegetation. Therefore, mountain vegetation is not considered in the present study. However, desert vegetation in the lower reaches of the Konqi River was included because vegetation growth depends on mountain-pass water.

## Data collection

### Land cover/use data

The present study collected Landsat TM remote-sensing images of August in 2013 with a resolution ratio of 30 m and the Digital Elevation Model (DEM) of the Kaidu-Konqi River Basin (Fig. 1). These images are of good quality and with no cloud interference. Geometric correction and the mosaic of the image were collected using ENVI 4.5 image-processing software (Harris Geospatial Solutions, Broomfield, CO, USA) based on 1;100,000 topographic maps, and WGS-1984-Albers Projection, which was selected for geometric correction. According to the research purpose and status of the study area, images were classified by using visual interpretation and supervised classification methods. The first grade includes six land-use types: water body, cultivated land, forestland, grassland, residential and industrial land, and unused land. Field observations clarified the presence of mainly six land use and land cover categories in the study area. The minimum level of interpretation accuracy in the identification of land use and land cover categories from remote sensor data was at least 87%. The area of the different land uses can be calculated by checking the attribute table and resolution ratio on land cover/use images. The sum of forestland area and grassland area is taken as the areas of natural oases, while the sum of residential and industrial land area, as well as cultivated land area, represent the area of artificial oases. There is no ice in this study area. Water includes rivers, Bosten Lake, and other small water systems. Unused land cannot be considered because a tract of unused land does not consume water resources in oases. In addition, 50 Digital Elevation Models (DEMs) covering the entire Xinjiang, ranging from N40°–44° and E82°–91°, were downloaded from http://www.gscloud.cn/. These DEMs were merged and cropped according to land cover/use image boundaries of the Kaidu-Konqi River Basin in order to obtain the Kaidu-Konqi River Basin DEM. Watersheds with more 100,000 m$^3$ flows in DEM were extracted by spatial analyst tools, including slope, aspect, fill, flow accumulation, flow direction, stream link, and watershed of Arcgis 10.2.2 (Esri, Redlands, CA, USA). Watershed boundaries, including the Yanqi Basin and the Konqi River Basin (the study area of this paper, Fig. 1), were manually extracted with Arcgis 10.2.2 (Esri, Redlands, CA, USA). Combined with land cover/use image and extracted watershed boundary images of the study area, land cover/use in the study area was registered by watershed boundary images.

### Available water resource quantity

In this study, total available water resource quantity is determined by the runoff of the Kaidu River and the Konqi River. The runoff data from 1958 to 2013 in the Kaidu River are derived from the Dashankou Hydrologic Station located in the mountain pass of the headstream of the Kaidu River (Fig. 1). Regarding the Konqi River, runoff data are from the Tashidian Hydrologic Station (Fig. 1) located in the region that flows out from Bosten Lake. Runoff data in the Konqi River from 1966 to 1974 are missing due to the influence of the Chinese Cultural Revolution.

In this study, using 2013 as the base year, 2025 and 2035 are the planning years. Development indicators in 2025 and 2035 refer to "Integrated Planning Report in Kaidu-Konqi River Basin" (*Xinjiang Uygur Autonomous Region water conservancy and*

*Hydropower Survey Design and Research Institute, 2012*). The specific ideas of social and economic development in the Kaidu-Konqi River Basin in 2025 and 2035 are to: (1) reasonably control agricultural development scale, optimize the agricultural structure, and promote agricultural industrialization in an all-round way. (2) accelerate the implementation of the conversion of advantageous resources, comprehensively promote the new industrialization and industrialization process. (3) focus on tourism and service industry, develop the third industry by leaps and bounds. (4) strengthen the construction of ecological environment protection and resource-friendly society.

Based on the specific ideas of social and economic development mentioned above, runoff data in 2025 and 2035 have been predicted and provided by the book titled "Comprehensive Planning in the Tarim River Basin in Xinjiang" (*Xinjiang Uygur Autonomous Region Water Conservancy and Hydropower Design and Research Institute, Ministry of Water Resources, 2010*; *Xinjiang Uygur Autonomous Region water conservancy and Hydropower Survey Design and Research Institute, 2012*). The runoff in 2025 and 2035 in the Kaidu River was set to $33.90 \times 10^8$ m$^3$ and $33.90 \times 10^8$ m$^3$, respectively, which is slightly less than normal flow. The runoff in 2025 and 2035 in the Konqi River was set to $15.55 \times 10^8$ m$^3$ and $12.79 \times 10^8$ m$^3$, respectively, both of which are slightly less than normal flow. The local government plans to control the quantity of available water resources in the Kaidu-Konqi River Basin to a normal flow level in the future in order to meet ecological, economic, and social needs for water resources in the study area.

### Current natural and artificial oases area

In the present study, the year of 2013 is regarded as the current year. Areas of natural oases, artificial oases, and bodies of water in the Yanqi Basin and the Konqi River Basin were obtained by DEM and land cover/use image in 2013. Water mainly includes rivers and lakes. However, when the water area in the Yanqi Basin was calculated, Bosten Lake was not included because the water evaporation of Bosten Lake is not affected by the quantity of available water resources in the Yanqi Basin. Therefore, the water area in the Yanqi Basin was calculated by the total water area provided by land cover/use images subtracted by the Bosten Lake water area of 988 km$^2$ in 2013.

In recent years, the increase in population increased the pressure on the environment, which intensified the conflict between animals and grassland overgrazing and led to grassland degradation. In addition, there is not enough water to supply desert riparian forests because of the contradiction between supply and demand of water resources, which results in the serious decay of vegetation. However, the areas of forest and grasslands cannot be reduced, and artificial oases can't be extended again in order to maintain the ecological balance, so it is assumed that the areas of natural oases, artificial oases, and water will be unchangeable in 2025 and 2035, and the same as in 2013.

### Groundwater depth

Groundwater depth and maximum groundwater depth can be utilized to calculate phreatic evaporation ($E_p$). They were collected by consulting the relevant literature (*Guo et al., 2013*). Nowadays, the local government and people have realized that it is very important to maintain a certain depth of groundwater, so they will take some

measures or policies to maintain the groundwater depth. Therefore, it is also assumed that the groundwater depth will be constant from 2013 to 2035.

### Meteorological data

The daily mean air pressure, wind speed, temperature, air relative humidity, sunshine hours, and the daily surface water evaporation of the 20-cm general evaporation dish ($E_{\Phi 20}$) from weather stations in Yanqi, Hejing, Heshuo, Bohu and Yuli counties, and Korla city were collected by the China Meteorological Science Data Sharing Service Network (http://data.cma.cn). Yanqi, Hejing, Heshuo, and Bohu counties were chosen as the Yanqi Basin. Korla city and Yuli County represent the Konqi River Basin. The daily $E_{\Phi 20}$ in 2013 in the Yanqi Basin and the Konqi River Basin was summed up, respectively.

The previous study has shown that the temperature of the oasis area of this basin increased at a rate of 0.167 °C/10a, and precipitation increased non-significantly (*Fu et al., 2013*). Therefore, temperature in 2025 and 2035 was calculated according to this rate. Other meteorological data in 2025 and 2035 was assumed to be the same as in 2013.

### Social-economy data

Population and industrial output need to be collected in this study. These data were obtained from *Chen (2014)*. Although the Konqi River Basin only occupies one part of Yuli county, most agricultural production and people's lives depend on the Konqi River, with very little use of water from the Tarim River. Social-economy data in Yuli County belong to the Konqi River Basin. In this study, the total social-economy data in Korla and Yuli County were used to represent the data in the Konqi River Basin. According to *Chen, Du & Chen, 2013* and the specific ideas of social and economic development mentioned in 'Available water resource quantity', social-economy data in 2025 and 2035 have been predicted.

## Methods

### Z index method

The *Z* index was utilized to analyze high- and low-flow variations for surface runoff within a certain period in the river basin (*Ling, Xu & Fu, 2013*). The annual runoff of the Kaidu River and the Konqi River for 1958–2013 follows a Person-III distribution, and the formula for calculating the *Z* index is:

$$Z = \frac{6}{C_s} \times (\frac{C_s}{2} \times \varphi_i + 1)^{1/3} - \frac{6}{C_s} + \frac{C_s}{6} \tag{1}$$

$$C_s = \frac{\sum_{i=1}^{n}(R_i - \bar{R})^3}{n \times \sigma^3} \tag{2}$$

$$\varphi_i = \frac{R_i - \bar{R}}{\sigma} \tag{3}$$

$$\sigma = \sqrt{\frac{1}{n-1}\sum_{i=1}^{n}(R_i - \bar{R})^2} \tag{4}$$

$$\bar{R} = \frac{1}{n} \times \sum_{i=1}^{n} R_i \tag{5}$$

where $Z$ is the $Z$ index; $C_s$ is the skewness coefficient; $\varphi_i$ is the standard variable; $R_i$ is the annual runoff; $n$ is the sample number; and $\bar{R}$ is the mean value of the annual runoff sequence.

According to the normal distribution curve of the $Z$ variable, the limiting value of the $Z$ index is divided into three grades and into high- and low-flow types (*Guo et al., 2016*).

### Natural vegetation ecological water demand

Natural vegetation ecological water demand was calculated using the area quota method. This method has been widely employed to study well-investigated areas. Although it is simple to use, the method requires a reasonable determination of the ecological water quota of different types of vegetation. Its calculation formula is:

$$W_p = \sum_{i=1}^{n} W_{pi} = \sum_{i=1}^{n} A_i m_{pi} \tag{6}$$

where $p$ is vegetation water demand guarantee rate; $A_i$ is the vegetation area of the type $i$; $m_{pi}$ is the water quota for the corresponding vegetation guaranteed rate; and $n$ is the number of vegetation types.

According to *Zhang et al. (2011)*, the water quota in the Yanqi Basin is 3,000 m$^3$ hm$^{-2}$ for woodland, 1,500 m$^3$ hm$^{-2}$ for sparse woodland ground (with less than half shrub land), and 2,250 m$^3$ hm$^{-2}$ for all lawn. Because the areas of different types of vegetation are difficult to obtain, the water quota of natural vegetation in the Yanqi Basin was considered as the average of that of the above three types of vegetation, i.e., 2,250 m$^3$ hm$^{-2}$, and the same as in 2025 and 2035.

### Water demand of artificial oasis

In artificial oases, water consumption of crops, residences, and industries were considered. Water consumption of residences and industries were calculated by using the same method, i.e., the quota method. Specifically, the populations of non-agricultural residents and agricultural residents, and the regional industrial output, respectively, multiply their respective daily water consumption quotas (*Fu et al., 2017*). According to the survey, daily water consumption of non-agricultural residents and agricultural residents was 239.59 L day$^{-1}$ per capita, 245.35 L day$^{-1}$ per capita and 257.86 L day$^{-1}$ per capita in 2013, 2025 and 2035 in the Yanqi Basin, and 59.90 L day$^{-1}$ per capita, 61.34 L day$^{-1}$ per capita and 64.46 L day$^{-1}$ per capita in 2013, 2025 and 2035 in the Konqi River Basin, respectively (*Chen, Du & Chen, 2013*). Water consumption of industrial production value per 10,000 yuan is 45 L. According to *Chen, Du & Chen (2013)*, non-agricultural residents, agricultural residents and industrial production value are $128.03 \times 10^4$, $51.08 \times 10^4$ and $1,066.15 \times 10^8$ yuan in 2025, and $180.35 \times 10^4$, $49.18 \times 10^4$ and $1,949.43 \times 10^8$ yuan in 2035.

Crop water consumption is one of the most important factors in the farmland water circulation system. Crop water consumption could be figured out based on measured soil moisture with the water balance method, or with the integrated climatological method. According to the formula recommended by the Food and Agriculture Organization of the United Nations, the formula of calculating crop water consumption is as follows, under

the conditional that soil moisture is not a limiting factor:

$$ET_p = K_c \times ET_0 \tag{7}$$

where, $ET_p$ is crop water requirement under adequate water supply (mm/d); $K_c$ is crop coefficient, which is obtained with adequate water supply test; the results recommended by literature were adopted in this paper, where the crop coefficient of spring wheat, corn and cotton (three crops with the largest planting area in the study area) is 0.85, 0.86 and 0.74 respectively. $ET_0$ is evapotranspiration of reference crop (mm/d), and Penman method was used to calculate $ET_0$.

$$ET_0 = \frac{0.408\Delta(R_n - G) + \gamma \frac{900}{T+273} U_2 VPD}{\Delta + \gamma(1 + 0.34U_2)} \tag{8}$$

In this formula, $ET_0$ is evapotranspiration of reference crop (mm/d), $R_n$ is surface net radiation of crops (MJ/m$^2$ d), $G$ is soil heat flux (MJ/m$^2$ d), $T$ is average temperature at 2 m height (°C), $U_2$ is 24 h mean wind speed at 2 m height (m/s), $VPD$ is vapor pressure deficit at 2 m height (kPa), $\Delta$ is saturation water vapor pressure slope (kPa/°C), $\gamma$ is dry and wet sphere constant (kPa/°C). $R_n$, $G$, $U_2$, $VPD$, $\Delta$ and $\gamma$ in 2013 are the same as in 2025 and 2035 because soil characteristics, regional atmospheric circulation and shape of evaporation surface will experience no major changes in the next 23 years, except in $T$. The previous study had shown that the temperature in this study area increased at a rate of 0.167 °C/10a (Fu et al., 2013). Therefore, $T$ in 2025 and 2035 were calculated at this rate. $ET_0$ in 2025 and 2035 were calculated using $T$ in 2025 and 2035. In crop growing areas, the growth of crops requires a lot of water. In addition, the soil salt discharge is also required for a certain amount of irrigation water. However, some scholars have found that the degree of salinization of cultivated land is far less than the wasteland (Ren et al., 2010). In this study, only cultivated land was considered. Moreover, the drip and pipe irrigation techniques have been used widely, which will reduce soil salinization in cultivated land. Therefore, water consumption of soil salt discharge has not been considered in this study.

### Model for calculating the most suitable oasis size

Based on the water balance in the Kaidu-Konqi River Basin, the following model was utilized for calculating the most suitable oasis size in arid regions (Lei, Li & Ling, 2015):

$$\frac{W - W_0}{(\alpha A_N + \beta A_A + A_W E_{\varphi 20} \gamma + A_0 E_p) 10^{-5}} = 1 \tag{9}$$

where $A_N$ is the area of a natural oasis (km$^2$); $A_A$ is the irrigated area of an artificial oasis (km$^2$); $A_A$ is the area of artificial water (km$^2$); $A_0$ is the area of residence site, and industrial area in the artificial oasis region (km$^2$); $W$ is the total quantity of available water resources ($10^8$ m$^3$); $W_0$ is non-vegetation water consumption, including industrial and domestic water uses, as well as surface evaporation and the minimum ecological water demand in the river channel ($10^8$ m$^3$); $\alpha$ and $\beta$ are the water demand quotas of the natural and artificial oases, which are 225 (Zhang et al., 2011) and 355 mm (this value was calculated by agricultural, non-agricultural residents and industrial output) (Chen, Du & Chen, 2013) in the Kaidu-Konqi River Basin, respectively; $E_{\varphi 20}$ is the surface water evaporation capacity

of a 20 cm general evaporation dish (mm); $E_p$ is the phreatic evaporation intensity (mm) from residence site and industrial area in an artificial oasis, calculated by formula (13). In the formula, the value of $H$ is 3.5 m in this river basin (*Guo et al., 2013*); and $\gamma$ is the conversion coefficient of surface evaporation, which has a value of 0.61 (*Zhou, 1999*).

### Optimal area of natural oases

The proportion of natural oasis to the total oasis is set as $\mu$.

$$A_N = \mu A \tag{10}$$

$$A_H = A_A + A_W + A_0 = (1 - \mu)A \tag{11}$$

where $A$ is the optimal area of a natural oasis in the study region ($km^2$); and $A_H$ is the optimal area of an artificial oasis ($km^2$). Equations (10) and (11) were put into Eq. (9), and the calculation model of the suitable oasis scale changes to:

$$A = \frac{(W - W_0)10^5 + A_w(\beta - E_{\varphi 20}\gamma) + A_0(\beta - E_p)}{\alpha\mu + \beta(1 - \mu)} \tag{12}$$

### Phreatic evaporation model

This study used a phreatic evaporation model to calculate evaporation from residence site and industrial area in an artificial oasis area as follows (*Ye, Chen & Li, 2007*):

$$E_p = a(1 - H/H_{\max})^b E_{\Phi 20} \tag{13}$$

where $E_p$ is the phreatic evaporation intensity from residence site and industrial area (mm); $E_{\Phi 20}$ is the surface water evaporation of the 20 cm general evaporation dish (mm); $H$ is the groundwater depth (mm); $H_{\max}$ is the critical phreatic water depth (m), which is 5 m in the study (*Guo et al., 2013*); $a$ and $b$ are the empirical coefficients and taken as $a = 0.62$ and $b = 2.8$ in these two regions, because of widely distributed meadow sand (*Shi, 2009*; *Ye, Chen & Li, 2007*).

## RESULTS

### High- and low-flow variations for surface runoff in the Kaidu-Konqi River

High- and low-flow variations for surface runoff were analyzed using the $Z$ index method (Fig. 2), based on annual runoff from the Kaidu-Konqi River from 1958 to 2013, as well as 2025 and 2035.

Figure 2A shows that the runoff in the Kaidu River Pass headstreams changed from a high- and normal-flow period into a low-flow period between 1958 and 1962. Four normal-flow years (1963–1966) occurred followed by two low-flow years (1967 and 1968). Two high-flow years (1970–1971) and 10 low-flow years (1974–1975, 1977–1979, 1981, 1983–1986) occurred between 1970 and 1989, and the rest were normal-flow periods. Runoff had a normal-flow period between 1990 and 1992, declined to a low-flow period in 1993 and 1995, and then increased to a high-flow period in 1994 and 1996. After 1996, the

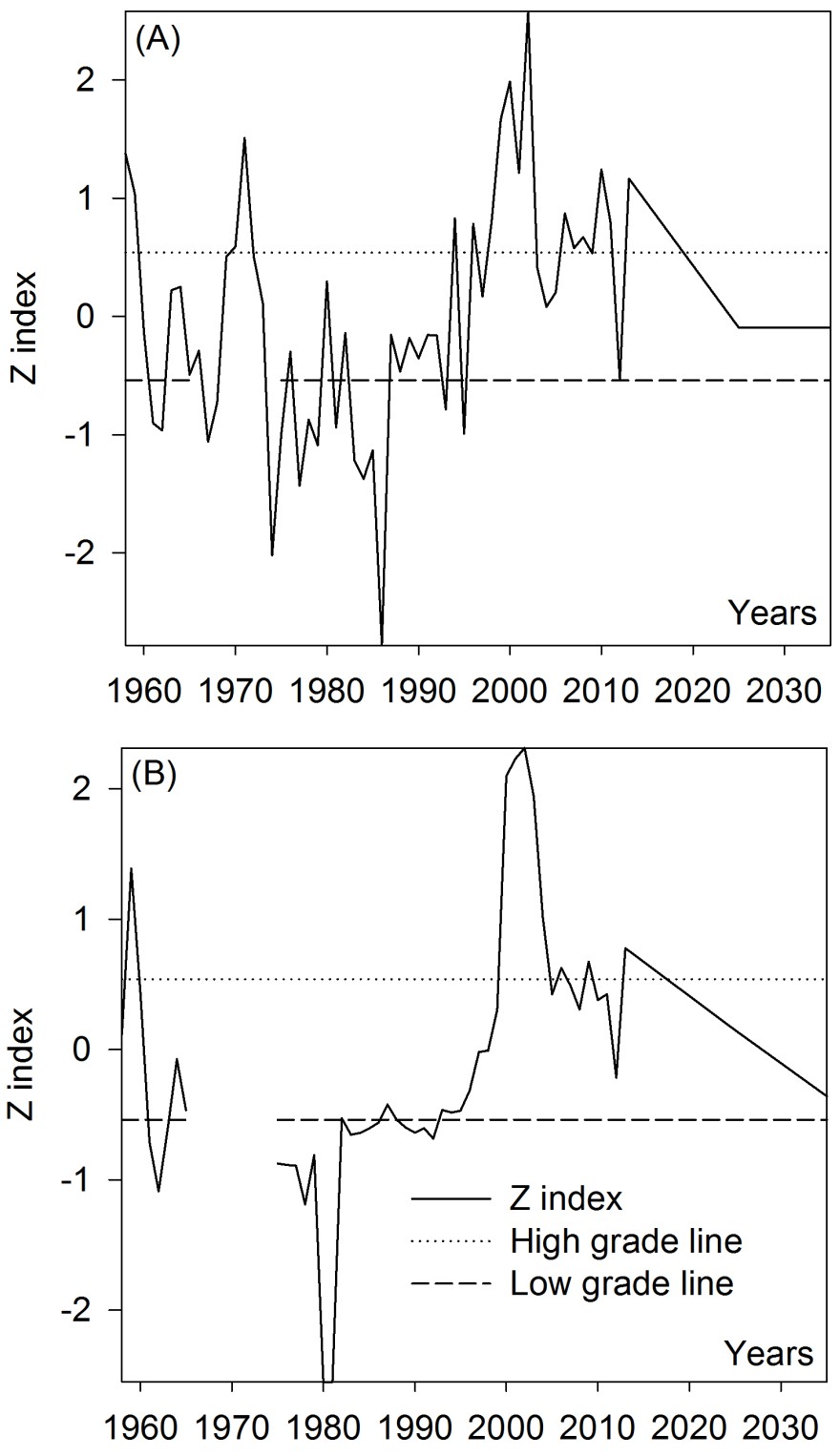

**Figure 2** **High-and low-flow variations of runoff in the Kaidu-Konqi River Basin.** Lines of 0.54 and −0.54 represent judgment boundary on grades of low-, normal- and high-flow index. (A) the Kaidu River (B) the Konqi River.

flows were mainly high, except 1997, 2003–2005, 2009 and 2012, which were normal-flow years.

From Fig. 2B, it can be seen that the runoff in the Konqi River had a normal-flow year in 1958, rose to a high-flow period in 1959, and then declined to a normal-flow year in 1960. After 1960, the flow entered a low-flow period between 1961 and 1963. Flows returned to a normal condition in 1964 and 1965. Flow conditions from 1966 to 1974 cannot be determined because of missing data. Runoff has generally been at a low-flow level since 1975, particularly between 1975–1981, 1983–1986, and 1989–1992, and the rest were normal-flow periods until 1999. Runoff has been at a high-flow level from 2000 to 2004, 2006, 2009 and 2013, and the rest were normal flow. The runoff of Kaidu and Konqi Rivers in 2025 and 2035 planned by the local government are all near the normal-flow level.

According to the above analysis, since the Kaidu River and the Konqi River have apparent high- and low-flow variations, it is feasible to determine suitable oasis sizes for the Kaidu-Konqi River Basin based on variations in surface runoff. The mean values for annual runoff in the three flow categories (high, normal, and low) between 1958 and 2013 in the Kaidu River and the Konqi River were $42.99 \times 10^8$ m$^3$, $34.48 \times 10^8$ m$^3$ and $28.91 \times 10^8$ m$^3$, as well as $22.25 \times 10^8$ m$^3$, $14.43 \times 10^8$ m$^3$ and $10.59 \times 10^8$ m$^3$, respectively. The local government has planned a runoff of $33.90 \times 10^8$ m$^3$ and $33.90 \times 10^8$ m$^3$ in the Kaidu River and a runoff of $15.55 \times 10^8$ m$^3$ and $12.79 \times 10^8$ m$^3$ in the Konqi River in 2025 and 2035 when the water inflow rate was 50%, respectively (*Xinjiang Uygur Autonomous Region water conservancy and Hydropower Survey Design and Research Institute, 2012*). In addition, the average runoff in the Kaidu River and the Konqi River was $35.05 \times 10^8$ m$^3$ and $13.34 \times 10^8$ m$^3$, respectively (*Guo et al., 2013*), and thus their minimum ecological base flow was $3.505 \times 10^8$ m$^3$ and $1.334 \times 10^8$ m$^3$ according to the Tennant method, respectively (*Ye, Chen & Li, 2007*). Moreover, the average available water quantity in the other water systems and groundwater in the Kaidu River and the Konqi River was $6.54 \times 10^8$ m$^3$ and $3.48 \times 10^8$ m$^3$ in 2013, respectively (*Xinjiang Uygur Autonomous Region water conservancy and Hydropower Survey Design and Research Institute, 2012*). The Kaidu River needs to provide $26.99 \times 10^8$ m$^3$ to Bosten Lake in order to maintain the ecological security of Bosten Lake (*Guo et al., 2013*). In addition, the Konqi River needs to provide $2.63 \times 10^8$ m$^3$ in 2013 and $4.50 \times 10^8$ m$^3$ in 2025 and 2035, respectively, to the Tarim River annually in order to ensure delivery of water to the lower reaches of the Tarim River (*Guo et al., 2013*). The total available water quantity of groundwater in the Konqi River planned by the government in 2025 and 2035 was $2.78 \times 10^8$ m$^3$ (*Xinjiang Uygur Autonomous Region water conservancy and Hydropower Survey Design and Research Institute, 2012*). Regarding the Kaidu River, the total available water quantity in groundwater and other water systems in 2025 and 2035 is unchangeable, and the same as in 2013. Accordingly, the total available water quantities after eliminating the river base flow and the input of water into Bosten Lake from the Kaidu River over the three periods, in 2025 and 2035 were $19.04 \times 10^8$ m$^3$, $10.52 \times 10^8$ m$^3$, $4.95 \times 10^8$ m$^3$, $9.95 \times 10^8$ m$^3$, and $9.95 \times 10^8$ m$^3$. The total available water quantities after eliminating the river base flow and the input of water into the Tarim

River in the Konqi River over the three periods, in 2025 and 2035 were $21.77 \times 10^8$ m$^3$, $13.95 \times 10^8$ m$^3$, $10.11 \times 10^8$ m$^3$, $12.50 \times 10^8$ m$^3$, and $9.74 \times 10^8$ m$^3$.

## Ecological water demand of a natural oasis

The natural oasis is impacted less by anthropogenic activities because it is located at the periphery of cultivated land and the lower reaches of the Konqi River Basin. The natural oasis mainly comprises forestland, shrub and grassland in this region. The natural oasis area in the Yanqi Basin and the Konqi River Basin was 1,151.51 km$^2$ and 2,041.09 km$^2$, respectively, based on Landsat TM remote-sensing images from 2013. Their water demand quota were all $22.5 \times 10^4$ m$^3$/km$^2$, and therefore their ecological water demand was $2.59 \times 10^8$ m$^3$ and $4.59 \times 10^8$ m$^3$, respectively. In addition, the water surface evaporation of the natural lake (Bosten Lake) was neglected because the evaporation capacity of Bosten Lake cannot be affected by the total quantity of available water resources in the Yanqi Basin. The area covered by the river channel and other water systems, except Bosten Lake, was 9.36 km$^2$ (997.36 km$^2$–988 km$^2$ of Bosten Lake in 2013) and 37.52 km$^2$, and water surface evaporation ($E_{\Phi 20}$) in the whole year of 2013 is 1,162.7 mm and 1,460.2 mm in the Yanqi Basin and the Konqi River Basin, respectively. Therefore, the total water consumption was $0.07 \times 10^8$ m$^3$ and $0.34 \times 10^8$ m$^3$ in these two regions, respectively, if the water surface conversion coefficient (0.61) is used. It was assumed that the ecological water demand of natural vegetation and the water surface evaporation of the river course are the same in 2025 and 2035, as in 2013. In the Yanqi Basin and the Konqi River Basin, if river course stability is assumed, then the available water resource quantities of oases in the Yanqi Basin in the high-flow, normal flow, and low-flow periods, 2025 and 2035 were $16.38 \times 10^8$ m$^3$, $7.86 \times 10^8$ m$^3$, $2.29 \times 10^8$ m$^3$, $7.29 \times 10^8$ m$^3$, and $7.29 \times 10^8$ m$^3$. The total available water quantities in the Konqi River over the three periods, 2025 and 2035 were $16.84 \times 10^8$ m$^3$, $9.02 \times 10^8$ m$^3$, $5.18 \times 10^8$ m$^3$, $7.57 \times 10^8$ m$^3$, and $4.81 \times 10^8$ m$^3$. It can be concluded that the remaining water resources, not including river base flow, river evaporation and inflow into Bosten Lake can meet the water needs of natural vegetation.

## Water demand of artificial oases

Land cover/use in 2013 showed that the cultivated area supplied by the artificial oasis in the Yanqi Basin and the Konqi River Basin was 1,848.52 km$^2$ and 1,794.68 km$^2$, respectively. The three crops with the largest planting area in this region are spring wheat, corn and cotton. The water consumptions of these three crops were calculated. The total water consumption of three crops was $10.46 \times 10^8$ m$^3$ in 2013, $10.94 \times 10^8$ m$^3$ in 2025 and $10.69 \times 10^8$ m$^3$ in 2035 in the Yanqi Basin, and $18.52 \times 10^8$ m$^3$ in 2013, $13.99 \times 10^8$ m$^3$ in 2025 and $13.21 \times 10^8$ m$^3$ in 2035 in the Konqi River Basin, respectively. Furthermore, the water consumption of the other land use types, such as residence site and industrial area, was $0.05 \times 10^8$ m$^3$ and $0.07 \times 10^8$ m$^3$ in these two regions, respectively, based on calculated phreatic evaporation intensity ($E_p$).

The agricultural and non-agricultural population were 301,100 and 197,700 in 2013, 232,900 and 292,500 in 2025, 207,700 and 356,000 in 2035 in the Yanqi Basin, and 270,600 and 409,300 in 2013, 277,900 and 987,800 in 2025, 284,100 and 1,447,500 in 2035 in the

Konqi River Basin, respectively, and daily water consumption of non-agricultural residents and agricultural residents was 239.59 L day$^{-1}$ per capita, 245.35 L day$^{-1}$ per capita and 257.86 L day$^{-1}$ per capita in 2013, 2025 and 2035 in the Yanqi Basin, and 59.90 L day$^{-1}$ per capita, 61.34 L day$^{-1}$ per capita and 64.46 L day$^{-1}$ per capita in 2013, 2025 and 2035 in the Konqi River Basin, respectively. Therefore, the population water consumption was $0.214 \times 10^8$ m$^3$, $0.282 \times 10^8$ m$^3$ and $0.345 \times 10^8$ m$^3$ in the Yanqi Basin, and $0.375 \times 10^8$ m$^3$, $0.851 \times 10^8$ m$^3$ and $1.287 \times 10^8$ m$^3$ in the Konqi River Basin in 2013, 2025 and 2035, respectively, if the water resource utilization coefficient was 0.9. Furthermore, the regional industrial output in the Yanqi Basin and the Konqi River Basin was $1,326,147 \times 10^4$ yuan and $6,600,403 \times 10^4$ yuan in 2013, $1,006,700 \times 10^4$ yuan and $9,654,800 \times 10^4$ yuan in 2025, and $1,855,300 \times 10^4$ yuan and $1,763,900 \times 10^4$ yuan in 2035, respectively. The water consumption quota for the industrial output of ten thousand yuan is 0.045 m$^3$, and thus the total industrial water consumption of the Yanqi Basin and the Konqi River Basin was $0.0006 \times 10^8$ m$^3$ and $0.003 \times 10^8$ m$^3$ in 2013, $0.0005 \times 10^8$ m$^3$ and $0.004 \times 10^8$ m$^3$ in 2025, and $0.0008 \times 10^8$ m$^3$ and $0.007 \times 10^8$ m$^3$ in 2035, respectively. According to the above calculations, the socioeconomic water consumption of the artificial oasis in these two regions was $0.2146 \times 10^8$ m$^3$ and $0.378 \times 10^8$ m$^3$ in 2013, $0.2825 \times 10^8$ m$^3$ and $0.855 \times 10^8$ m$^3$ in 2025, and $0.3458 \times 10^8$ m$^3$ and $1.294 \times 10^8$ m$^3$ in 2035, respectively, and the total water consumption of the artificial oasis was $10.67 \times 10^8$ m$^3$ in 2013, $11.22 \times 10^8$ m$^3$ in 2025 and $11.04 \times 10^8$ m$^3$ in 2035 in the Yanqi Basin and $18.90 \times 10^8$ m$^3$ in 2013, $14.85 \times 10^8$ m$^3$ in 2025, and $14.50 \times 10^8$ m$^3$ in 2035 in the Konqi River Basin, which were more than $9.95 \times 10^8$ m$^3$and $12.12 \times 10^8$ m$^3$ calculated by *Chen, Du & Chen (2013)*. This is because the cultivated area in the Kaidu-Konqi River Basin predicted by *Chen, Du & Chen (2013)* was only 2023 km$^2$ in 2015, which is less than the 3,643.20 km$^2$ derived from satellite images in 2013. Therefore, the results predicted by *Chen, Du & Chen (2013)* were smaller than the results in this paper. Therefore, the respective available water resource quantity of the Yanqi Basin was $5.71 \times 10^8$ m$^3$, $-2.81 \times 10^8$ m$^3$, $-8.38 \times 10^8$ m$^3$, $-3.93 \times 10^8$ m$^3$ and $-3.75 \times 10^8$ m$^3$ for the high-, normal-, low-flow periods, in 2025 and 2035. The water shortage quantity of the Konqi River Basin was $-2.06 \times 10^8$ m$^3$, $-9.88 \times 10^8$ m$^3$, $-13.72 \times 10^8$ m$^3$, $-7.28 \times 10^8$ m$^3$ and $-9.69 \times 10^8$ m$^3$ for the high-, normal-, and low-flow periods, in 2025 and 2035. It can be concluded that, at present, available water resources not including river base flow, river evaporation and water requirement of natural oases, cannot meet the needs of artificial oases for water. If the cultivated area is not reduced in 2025 and 2035, water resource planning by the local government cannot meet the needs of artificial oases, and thus expansion or reducing the area of the artificial oases and determining the suitable development scale of oases are currently critical.

## Optimal size for an oasis under different inflow variations

Table 1 shows that the optimal size of the natural oasis in the Yanqi Basin and the Konqi River Basin should not be below 1,996.83 km$^2$ and over 7,301.11 km$^2$, respectively. If the normal flow period (23 years in the Yanqi Basin and 29 years in the Konqi River Basin) and low-flow period (16 years in the Yanqi Basin and 18 years in the Konqi River Basin) occupied 70% and 84% of the last 56 years, then the optimal area of the natural oasis in

**Table 1** Areas of natural oases and total oases in the Yanqi Basin and the Konqi River Basin.

| Basin | Periods of the different grade | Water resource quantity ($10^8$ m$^3$) | Natural oasis ($A_N$) (km$^2$) | | Artificial oasis ($A_H$) (km$^2$) | | Total oases ($A$) (km$^2$) | |
|---|---|---|---|---|---|---|---|---|
| | | | Suitable scale | Actual scale | Suitable scale | Actual scale | Suitable scale | Actual scale |
| Yanqi Basin | High-flow period | 19.04 | 7,301.11 | 1,151.51 | 2,282.13 | 1,848.52 | 9,583.24 | 3,000.03 |
| | Normal-flow period | 10.52 | 4,093.70 | | 1,279.58 | | 5,373.28 | |
| | Low-flow period | 4.95 | 1,996.83 | | 624.15 | | 2,620.99 | |
| | 2025 | 9.95 | 3,867.57 | | 1,232.49 | | 5,100.06 | |
| | 2035 | 9.95 | 3,872.92 | | 1,223.22 | | 5,096.15 | |
| Konqi River Basin | High-flow period | 21.77 | 8,726.21 | 2,041.09 | 1,761.34 | 1,794.68 | 10,487.55 | 3,835.77 |
| | Normal-flow period | 13.95 | 5,613.64 | | 1,133.09 | | 6,746.73 | |
| | Low-flow period | 10.11 | 4,085.22 | | 824.58 | | 4,909.81 | |
| | 2025 | 12.50 | 5,097.50 | | 911.03 | | 6,008.53 | |
| | 2035 | 9.74 | 3,994.68 | | 696.68 | | 4,691.36 | |

**Notes.**

$A_N$ is the area of a natural oasis (km$^2$); $A_H$ is the optimal area of an artificial oasis (km$^2$); $A$ is the optimal area of an oasis in the study area (km$^2$).

these two regions is 1,996.83–4,093.70 km$^2$ and 4,085.22–8,726.21 km$^2$, and the optimal area covered by all the oases should be 2,620.99–9,583.24 km$^2$ and 4,909.81–10,487.55 km$^2$, respectively. The actual natural oasis area of these two regions in 2013 was 1,151.51 km$^2$ and 2,041.09 km$^2$ less than the suitable scale, and their ecological water demand was $2.59 \times 10^8$ m$^3$ and $4.59 \times 10^8$ m$^3$, respectively. Therefore, high-normal-low flow can meet the needs of natural oases for water resources in the study area.

As shown in Table 1, the suitable scales of artificial oases in the Yanqi Basin during normal and low flow periods are lower than the actual area, which indicates that the artificial oasis area needs to be decreased by 568.94–1,224.37 km$^2$. Nevertheless, the artificial oasis area needs to be decreased rapidly by 133.34–970.10 km$^2$ in high-, low-, and normal-flow runoff due to the severe shortage of water resources in the Konqi River Basin.

The local government has planned the quantity of available water resources in the Kaidu-Konqi River Basin to be slightly less than the normal-flow level in 2025 and 2035 (Table 1). It has been calculated that the suitable scale of the natural oasis is 3,867.57 km$^2$ and 3,872.92 km$^2$ in 2025 and 2035, respectively, in the Yanqi Basin, and 5097.50 km$^2$ in 2025 and 3,994.68 km$^2$ in 2035, respectively, in the Konqi River Basin. Compared to the actual scale of the natural oasis in 2013, the natural oasis needs to be increased by 2,721.41 km$^2$ and 1,953.59 km$^2$ till 2035 in the Yanqi Basin and the Konqi River Basin, respectively, by returning farmland to forest and restoring desert vegetation. Compared to the actual scale of the artificial oasis in 2013, the artificial oasis needs to be decreased by 616.03 km$^2$ in 2025 and 625.30 km$^2$ in 2035 in the Yanqi Basin and 883.65 km$^2$ in 2025 and 1,097.99 km$^2$ in 2035 in the Konqi River Basin, respectively.

## Calculating suitable sizes of oases under appropriate proportioning

*Hu et al. (2006)* suggested that the area of a natural oasis should comprise 60% of the total oases area in any given arid region. *Lei, Li & Ling (2015)* also reported that the proportion of land occupied by natural and artificial oases (i.e., natural oasis occupies 60%, and

**Table 2  The optimal areas of artificial and natural oases under appropriate proportioning.**

| Basin | Periods of the different grade | $W - W_0$ $(10^8 \text{ m}^3)$ | $A_W$ $(\text{km}^2)$ | $A_0$ $(\text{km}^2)$ | $\mu$ (%) | $E_p$ (mm) | Suitable scale $(\text{km}^2)$ | | |
|---|---|---|---|---|---|---|---|---|---|
| | | | | | | | Total oases (A) $(\text{km}^2)$ | Natural oasis $(A_N)$ $(\text{km}^2)$ | Artificial oasis $(A_H)$ $(\text{km}^2)$ |
| Yanqi Basin | High-flow period | 18.99 | 9.36 | 123.23 | 0.77 | 24.76 | 9,345.30 | 7,626.72 | 1,718.58 |
| | Normal-flow period | 10.47 | | | 0.77 | | 5,239.86 | 4,276.26 | 963.60 |
| | Low-flow period | 4.90 | | | 0.77 | | 2,555.91 | 2,085.88 | 470.03 |
| | 2025 | 9.90 | | | 0.77 | | 4,972.71 | 4,041.85 | 930.86 |
| | 2035 | 9.90 | | | 0.77 | | 4,969.22 | 4,046.61 | 922.61 |
| Konqi River Basin | High-flow period | 21.70 | 37.52 | 119.65 | 0.82 | 31.10 | 10,425.40 | 8,811.25 | 1,614.16 |
| | Normal-flow period | 13.88 | | | 0.82 | | 6,706.75 | 5,668.35 | 1,038.40 |
| | Low-flow period | 10.04 | | | 0.82 | | 4,880.71 | 4,125.04 | 755.68 |
| | 2025 | 12.43 | | | 0.84 | | 5,975.17 | 5,143.15 | 832.02 |
| | 2035 | 9.67 | | | 0.84 | | 4,665.67 | 4,029.83 | 635.85 |

**Notes.**
$A_W$ is the area of artificial water $(\text{km}^2)$; $A_N$ is the area of a natural oasis $(\text{km}^2)$; $A_H$ is the optimal area of an artificial oasis $(\text{km}^2)$; $A$ is the optimal area of an oasis in the study area $(\text{km}^2)$; $A_0$ is the area of residence site, and industrial area in the artificial oasis region $(\text{km}^2)$; $W$ is the total quantity of available water resources $(10^8 \text{ m}^3)$; $W_0$ is non-vegetation water consumption, including industrial and domestic water uses, as well as surface evaporation and the minimum ecological water demand in the river channel $(10^8 \text{ m}^3)$; $E_p$ is the phreatic evaporation (mm) from construction land in an artificial oasis; $\mu$ is the proportion of natural oasis to the total oasis.

artificial oasis occupies 40%) is feasible in the Keriya River Basin. After the area of artificial oasis and natural oasis in 2013 is redistributed according to 6:4 ratio in this study area, the optimal areas of natural and artificial oases were re-calculated by using the area of redistributed natural oasis based on the constructed mathematical model (Eq. (10)) in this paper (Table 2).

From Table 2, it can be seen that compared to 38% of the natural oasis occupation rate in the Yanqi Basin (*Guo et al., 2016*), the suitable scales of natural oasis calculated according to 60% of the oasis occupation rate increased by 89.05–325.61 km² and suitable scales of artificial oasis decreased by 154.13–563.55 km². Thus, the suitable scales of total oasis decreased by 65.07–237.94 km², which indicates that natural oasis needs to be increased slightly, and artificial oasis needs to be decreased in order to improve the natural oasis occupation rate to 60%. In 2025 and 2035, the suitable scale of natural oasis increased by 174.28 km² and 173.69 km², but the artificial oasis suitable scale decreased by 301.63 km² and 300.62 km². Thus, the suitable scale of the total oases decreased by 127.36 km² and 126.93 km².

In the Konqi River Basin, when the natural oasis occupation rate changes from 53% to 60%, the suitable scale of natural oasis only needs to be increased by 54.71–85.04 km² in high-, normal-, and low-flow periods. Artificial oases decreased by 94.68–147.18 km², and thus the suitable scale of total oases decreased by 39.98–62.14 km², which indicates that the scale of the total oases changes non-significantly when the natural oasis occupation rate changes from 53% to 60%. This is likely because the natural oasis occupation rate at present in the Konqi River Basin is 53%, which is closer to 60% than 38% in the Yanqi Basin. Therefore, the change of the oasis scale in the Konqi River Basin is not obvious as the natural oasis occupation rate ranging from 53% to 60% than in the Yanqi Basin. The

suitable scale of natural oases increased by 35.15 km$^2$ and 45.65 km$^2$ in 2025 and 2035, respectively, and the suitable scale of artificial oasis decreased by 60.84 km$^2$ and 79.00 km$^2$, respectively. Thus, the suitable scale of the total oasis decreased by 33.36 km$^2$ and 25.69 km$^2$ in 2025 and 2035, respectively.

## DISCUSSION

According to *Li, Feng & Guo (2008)*, oases of 90% have no appropriate proportion of artificial and natural oases in arid regions of China, and artificial oases covering 59% cannot be maintained. At present, artificial oases in the Yanqi Basin and the Konqi River Basin comprise 62% and 47% of the total oases, respectively. They are all more than 40%, and these oases cannot be maintained. For this reason, the determinants of suitable oases sizes should be further investigated. There have been few studies investigating the optimal sizes of oases in arid regions. Some studies have focused on the optimal size of artificial oases (*Hu et al., 2006*; *Huang, Shen & Zhang, 2008*; *Li et al., 2011*) and natural oases (*Lei, Li & Ling, 2015*; *Guo et al., 2016*). A few studies have considered regional water resource changes and the ecological water demand of natural oases (*Lei, Li & Ling, 2015*; *Guo et al., 2016*). However, these studies only calculated the suitable oases scales at present, and did not consider suitable size changes in the future under government planning for water resource allocation. Therefore, this paper utilized GIS technology and field survey techniques to analyze the changes of water resources, and natural and artificial oases, and confirmed the optimal size of natural and artificial oases at present and in 2025 and 2035 impacted by the local government planning.

As in *Lei, Li & Ling (2015)* and *Guo et al. (2016)*, this paper regarded the change in water resources as the primary factor because total water quantity determines the development scale of oases in arid regions (*Li et al., 2011*). In the current study, runoff variation in the past 56 years in the Yanqi Basin and Konqi River Basin was analyzed, and it was found that runoff was mainly in normal- and low-flow periods. It was also determined that the available water quantity cannot meet the water demand of artificial oases when water requirement of the natural oasis was satisfied in the Yanqi Basin and the Konqi River Basin. Therefore, compared with 2013, the artificial oasis in the Yanqi Basin needs to be decreased by 616.03 km$^2$ and 625.30 km$^2$ in 2025 and 2035, respectively. The Konqi River Basin needs to be reduced by 883.65 km$^2$ and 1,097.99 km$^2$ in 2025 and 2035, respectively, compared with 2013. In addition, the decrease scope of the suitable artificial oasis sizes in this region is much more than 487 km$^2$ (the decrease scope of the entire suitable oasis sizes) of the Keriya River Basin (*Lei, Li & Ling, 2015*) and 148 km$^2$ (the decrease scope of the entire suitable oasis sizes) of the Hotan River Basin (*Guo et al., 2016*).

The optimal proportion of the natural to artificial oases was 6:4 (i.e., natural oasis occupies 60% of the whole oasis, and the artificial oasis is 40%). Natural oasis did not occupy 60% of the whole oasis in the Kaidu-Konqi River Basin in 2013, i.e., only 38% and 53%. In this paper, the suitable oases scale was calculated again after the area of artificial oasis and natural oasis in 2013 is redistributed according to 6:4 ratio. The result showed that the natural oases area needs to be increased slightly compared to the proportion of

original oases, whereas the artificial oases area needs to be decreased to a greater extent. Therefore, the deficient natural oases area and redundant artificial oases area at present cannot maintain a healthy ecosystem because consuming too much water for artificial oases would seriously crowd-out ecological water leading to serious water shortage. To solve this problem, the local government has decided to adjust available surface water and the amount of groundwater exploitation, and established that the total quantity of available water resources in 2025 and 2035 would be slightly less than that of the normal-flow period in 2013 in the Yanqi Basin and the Konqi River Basin. Based on these planned data, suitable oasis sizes were calculated, and it was found that the natural oasis scale needs to be increased in the future, but not artificial oasis scales in the Yanqi Basin and the Konqi River Basin compared to 2013.

## CONCLUSIONS

The Yanqi Basin and Konqi River Basin, which are in an extremely arid region of China, were chosen as the study area. Based on water resources, weather, socio-economic and remote sensing image data, and government plan reports, suitable scales of oases under different high-, low- and normal-flow variations and the government plan for available water resources were analyzed by using the $Z$ index and the water balance method. The following conclusions were obtained:

(1) The Kaidu River and the Konqi River have high- and low-flow variations. The local government has planned water resources quantity in 2025 and 2035. The total available water quantities after eliminating the river base flow and the input of water into Bosten Lake and the Tarim River in the Kaidu-Konqi River over the three periods, in 2025 and 2035 were $19.04 \times 10^8$ m$^3$, $10.52 \times 10^8$ m$^3$, $4.95 \times 10^8$ m$^3$, $9.95 \times 10^8$ m$^3$ and $9.95 \times 10^8$ m$^3$, as well as $21.77 \times 10^8$ m$^3$, $13.95 \times 10^8$ m$^3$, $10.11 \times 10^8$ m$^3$, $12.50 \times 10^8$ m$^3$, and $9.74 \times 10^8$ m$^3$.

(2) The ecological water demand of the natural oases was $2.59 \times 10^8$ m$^3$ and $4.59 \times 10^8$ m$^3$ in the Yanqi Basin and the Konqi River Basin, respectively. The respective development scales in the natural oasis during the high-, normal- and low-flow periods, in 2025 and 2035 were 7,301.11 km$^2$, 4,093.70 km$^2$, 1,996.83 km$^2$, 3,867.57 km$^2$ and 3,872.92 km$^2$ in the Yanqi Basin, and 8,726.21 km$^2$, 5,613.64 km$^2$, 4,085.22 km$^2$, 5,097.50 km$^2$ and 3,994.68 km$^2$ in the Konqi River Basin. The suitable scales in high-, normal- and low-flow periods in the Yanqi Basin and Konqi River Basin were all more than 1,151.51 km$^2$ and 2,041.09 km$^2$, respectively, of the actual oasis areas. Therefore, the runoff among high-, normal- and low-flow not only can meet water demand of current natural oasis, but can support 845.32–6,149.60 km$^2$ in the Yanqi Basin and 2,044.13–6,685.12 km$^2$ in the Konqi River Basin of natural oasis.

(3) The total water consumption of the artificial oasis was $10.67 \times 10^8$ m$^3$, $11.22 \times 10^8$ m$^3$ and $11.04 \times 10^8$ m$^3$ in the Yanqi Basin, and $18.90 \times 10^8$ m$^3$, $14.85 \times 10^8$ m$^3$ and $14.50 \times 10^8$ m$^3$ in the Konqi River Basin in 2013, 2025, and 2035. The total available water cannot meet the needs of the artificial oasis in the Yanqi Basin and the Konqi River Basin after satisfying water demand of natural oases. The respective development

scales of the artificial oases during the high-, normal- and low-flow periods, in 2025 and 2035 were 2,282.13 km$^2$, 1,279.58 km$^2$, 624.15 km$^2$, 1,232.49 km$^2$ and 1,223.22 km$^2$ in the Yanqi Basin, and 1,761.34 km$^2$, 1,133.09 km$^2$, 824.58 km$^2$, 911.03 km$^2$ and 696.68 km$^2$ in the Konqi River Basin. The suitable artificial oasis area calculated in high-flow period was a little greater than 1,848.52 km$^2$ of the actual oasis area, other suitable scales calculated were less than 1,848.52 km$^2$ and 1,794.68 km$^2$ of the actual oasis area, respectively, which indicated that artificial oases in the Yanqi Basin and the Konqi River Basin need to be reduced more in the future.

(4) Using 6:4 as the proportion of the natural to artificial oases, the size of the natural oasis in the Yanqi Basin and the Konqi River Basin should be 2,085.88–7,626.72 km$^2$ and 4,125.04–8,811.25 km$^2$, respectively, and the size of the artificial oasis should be 470.23–1,718.58 km$^2$ and 755.68–1,614.16 km$^2$, respectively. The suitable total oases scale under the 6:4 proportion of the natural to artificial oases in the Yanqi Basin and the Konqi River Basin will decrease by 65.08–237.94 km$^2$ and 29.09–62.14 km$^2$, respectively, in 2025 and 2035, compared to the original proportion. However, the suitable scale of artificial oases will decrease by 300.62 km$^2$ and 60.84 km$^2$ until 2035 in the Yanqi Basin and the Konqi River Basin, respectively.

(5) The artificial oases in the Yanqi Basin and the Konqi River Basin tend to decrease due to limited water resources. Consequently, a substantial amount of farmland should be returned to forestland or grassland in these basins.

### Funding

This work was supported by the Strategic Priority Research Program of Chinese Academy of Sciences (No. XDA20100300) and Key Projects of National Natural Science Foundation of China (No. 41630859). The funders had no role in study design, data collection and analysis, decision to publish, or preparation of the manuscript.

### Grant Disclosures

The following grant information was disclosed by the authors:
Strategic Priority Research Program of Chinese Academy of Sciences: XDA20100300.
Key Projects of National Natural Science Foundation of China: 41630859.

### Competing Interests

The authors declare there are no competing interests.

### Author Contributions

- Aihong Fu analyzed the data, prepared figures and/or tables, authored or reviewed drafts of the paper, approved the final draft.
- Weihong Li contributed reagents/materials/analysis tools.
- Yaning Chen conceived and designed the experiments.
- Yuting Liu performed the experiments, helped draw Figure 1.

## Data Availability

The raw data are provided as Data S1.

## Supplemental Information

Supplemental information for this article can be found online at http://dx.doi.org/10.7717/peerj.4943#supplemental-information.

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
