# Peer review of "Suitable oasis scales under a government plan in the Kaidu-Konqi River Basin of northwest arid region, China"

_PeerJ, doi:10.7717/peerj.4943_

## Round 0.1 · original submission · Major Revisions

Dear Aihong,

I have received reviewers’ comments.

Two reviewers both raised key issues of experiment design and equations 6 and 7. Several inconsistences (for example in lines 125, 259, 475) were also identified.

Please make revision of the ms according to reviewers comment. I look forward to see your revised version.

Best wishes,

Hong YANG

Reviewer 1 ·

Basic reporting

1、 line108—109: Temperature, precipitation and solar shortwave radiation are collected, but no contribution to the study is seen in the paper.
2、 Line109: Precision of data fusion ALOS with ETM should be 2.5m, not 30m?
3、 Line 123-131:The river basin and its relationships are not clear,the water balance is convenient to be calculated. The color of the map is not standardized.
4、 Line 125:The total area of the Kaidu-Konqi River Basin is different Line 90 , line 125-126.
5、 Line158: In figure 1, recommend you supplementary main river system and basin boundary; adjust color of various area; “Kaidu River Basin” doesn’t exhibit; account for empty patch in main figure; Scale unit adjusted to km.
6、 Line154: The water balance calculation in the study area is based on the land use data, but the imaging time, pretreatment process, classification method and classification accuracy of the remote sensing image are not explained.
7、 line156—157: 109 lines refer to ALOS data , 382 lines refer to the use of ALOS to obtain natural oasis area ( land use ) , which is 2.5 m after ALOS data fusion , but the 30m data referred to here is used to obtain land use . please account for The data sources of land use .
8、 Line164: The highest version of ArcGIS is 10.5, currently.

Experimental design

9、 P168—170 the land use classification of USGS is simply introduced. How to merge it into the natural oasis and the artificial oasis?
10、 line171 : Since the study area is not of this type , the types of land use should be 6 .
11、P175 please replenish the information of spatial data projection.For the study area, the area data is very important. Based on the area error, the projection of spatial data is reconsidered.
11、 line190: The relationship between Kaidu - Konqi River Basin , Yanqi Basin and Konqi River Basin is unclear in the discussion of Figure 1 , and is suggested to be identified in Figure 1 .
12、 P204-205 This assumption is not seem realistic
13、 Line 229-230:In the calculation of hydraulic balance in 2020 and 2030, the area of oasis, groundwater depth, ,social economic data, the natural vegetation water demand, artificial oasis water demand as a constant,why?
15. P228-229 The size of artificial oasis mostly depends on socioeconomic data. We should emphasize human activity factors. Assuming that 2013-2030 years of socio-economic data remain unchanged, it means to weaken human activities .
16 lineP254—255 The ecological water demand quota method is too simplified for ecological water demand , and the result maybe not determined . It is suggested that crop transpiration water consumption model should be used to calculate the ecological water demand .
17、lineP257: what is mean of each letter in the formula?

Validity of the findings

18、line273-274: Artificial oasis uses unified quota calculation, ignoring the difference of different crops, increasing the uncertainty of the results. According to the relevant studies, the salinization phenomenon is more prominent in the study area, the local government is used to irrigate and
19、line427-429:why is the results of remote sensing images much larger than the results of Chen, please explain.
20、increase the water consumption of Salt discharge.

Reviewer 2 ·

Basic reporting

please see the General comments for the author.

Experimental design

please see the General comments for the author.

Validity of the findings

please see the General comments for the author.

Additional comments

The writting of this paper is very poor. There are many errors.
The methods to determine the suitable or optimal oasis area do not have robust physical base.
I don't think the current form shows contributions to the community as the methods are not novel and the results can not supported by the data.

1.The equation (7) is wrong, please carefully check the wording in the citation, especially the location of this item “10-5”. I do not agree with this formula, the reason is as follows: “Ep is the phreatic evaporation from construction land in an artificial oasis”, as I know the phreatic evaporation you used in your text (equation (11)) is the vegetation water use, not the construction land water use. And you had just classified the land use/cover into six types including the forest land, grassland, water, residence site, cultivated land and unused land. Did you mean that the construction land was the residence site? You also mentioned that “AA is the area of an artificial oasis”, however, I thought AA was the area of cultivated land, where you had mentioned above that artificial land included cultivated land and residence site, and you had already omitted the industrial and domestic water uses in the numerator of the equation (7).

2.Line 15, “and weather and socioeconomic data, suitable scales of oases were analyzed.” However, I did not the find the detailed weather data, except the introduction in the study region. Please add one figure about the detailed weather data, or just rewrite this sentence.

3.Line 251, The equation (6) is wrong, please add ∑ in the right side of the formula, and check carefully.

4.Line 259, the water quota of natural vegetation was 2250 m3 hm-2, therefore, the water quota was 225 mm, however, in Line 287, the water demand quota of the natural oases was 400 mm. Please unify the water quota. Similarly, water demand quota of the artificial oases was not unified.

5.Line 302, In the cited reference, the A was the optimal area of a natural oasis, however, in your text, you had changed to the optimal area of an oasis, and you used the same equation as the cited reference. Please check it carefully.

6.Line 475, “natural oasis occupies 60%, and artificial oasis occupies 40%”, however, the proportion of natural oasis and artificial oasis in Table 2 was not the rate.

---

## Round 0.2 · accepted · Accept

You have made changes to meet the requirements of reviewers.

Reviewer 2 ·

Basic reporting

please see the comments for the author.

Experimental design

please see the comments for the author.

Validity of the findings

please see the comments for the author.

Additional comments

The revised version improved much. All my comments have been addressed well.